# From *Contagium vivum fluidum* to *Riboviria*: A Tobacco Mosaic Virus-Centric History of Virus Taxonomy

**DOI:** 10.3390/biom12101363

**Published:** 2022-09-24

**Authors:** F. Murilo Zerbini, Elliot W. Kitajima

**Affiliations:** 1Department of Plant Pathology, Federal University of Viçosa, Viçosa 36570-900, MG, Brazil; 2Department of Plant Pathology and Nematology, University of São Paulo, Piracicaba 13418-900, SP, Brazil

**Keywords:** tobacco mosaic virus, TMV, virus taxonomy, Ivanovsky

## Abstract

Viruses were discovered as agents of disease in the late 19th century, but it was not until the 1930s that the nature of these agents was elucidated. Nevertheless, as soon as viral diseases started to be recognized and cataloged, there were attempts to classify and name viruses. Although these early attempts failed to be adopted by the nascent virology community, they are evidence of the human compulsion to try to organize the natural world into well-defined categories. Different classification schemes were proposed during the 20th century, but again none were widely embraced by virologists. In 1966, with the creation of the International Committee on Nomenclature of Viruses (eventually renamed as the International Committee on Taxonomy of Viruses), a more organized effort led to an official taxonomy in which viruses were classified into families and genera. At present, a much better understanding of the evolutionary relationships among viruses has led to the establishment of a 15-rank taxonomy based primarily on these evolutionary relationships. This review of virus taxonomy will be centered on the tobacco mosaic virus (TMV), the agent of the disease studied by Dmitry Ivanovsky and the first virus to be recognized as such, which was often historically at the center of major advancements in virology during the 20th century.

## 1. Introduction

Virology is a young field. A little more than 100 years have passed since the discovery of viruses as agents of disease. It was the late 19th century, and tobacco (a New World plant that had been introduced into Europe during the Columbian exchange [1]) was widely consumed on that continent. As with many introduced crops which came to be cultivated on a large scale outside their centers of origin, tobacco became afflicted by a number of diseases of unknown cause, including the “mosaic” disease. At that time, plant diseases were known to be caused by fungi, and whether they could also be caused by bacteria was a matter of debate [2]. Nothing smaller than bacteria was known to science at that time.

It was in this context that three European scientists started investigating the cause of tobacco mosaic disease. In the Netherlands, Adolf Mayer injected healthy tobacco plants with sap prepared from diseased plants and reported that the injected plants developed the exact same disease [3]. Thus, the infectious nature of tobacco mosaic was established. Soon after, Dmitry Ivanovsky passed the sap from infected tobacco plants through porcelain filters (“Chamberland” filters), a recently developed technology that was capable of retaining bacteria. Astonishingly, healthy tobacco plants injected with the filtered sap developed mosaic disease [4]. At almost the same time, the same experiment was performed by Martinus Beijerinck in the Netherlands, with the same results [5]. Beijerinck concluded that the causal agent of tobacco mosaic was not a fungus or a bacterium, but rather something entirely different and capable of passing through the pores of a Chamberland filter. He called the agent a *contagium vivum fluidum*, and maybe because this was too long, he “abbreviated” it as virus (a Latin word that means poison) [5]. Henceforth, the agent of tobacco mosaic disease was called the tobacco mosaic virus, and a new field of science was born.

This beautiful origin story, which can be found in every plant virology textbook, would not be complete without mentioning the fact that calling the tobacco mosaic virus a *contagium vivum fluidum* is not factually correct. It is certainly contagious (*contagium*), and the vast majority of virologists will now agree that it is also alive (*vivum*). However, viruses are not fluids (*fluidum*). At about the same time, Friedrich Loeffler and Paul Frosch, working on foot-and-mouth disease, not only demonstrated that the agent was infectious, but also proposed its nature to be particulate matter. They came to this conclusion using another newly developed technology called fine-grain Kitasato filtration, after Chamberland filtration [6]. Although infectivity was maintained after passing through the Chamberland filter, it was lost after passing through the less-porous Kitasato filter. Thus, the concept proposed by Loeffler and Frosch, of a new class of particulate infectious agents, was certainly more correct.

## 2. The Early Years of Virology and Initial Attempts at Classifying Viruses

Taxonomy, involving both classification and nomenclature, is a human compulsion. We (or at least some of us) have a primal need to organize the natural world into discrete categories based on a specific set of criteria (classification) and then name these categories and the organisms that are classified into them (nomenclature). By the time that viruses were unveiled as agents of disease in humans, other animals, and plants, a taxonomic framework already existed for animals and plants, based on the work of the Swedish scientist Carl Linnaeus (1707–1778). Obviously, the set of criteria that were used to define each taxonomic rank varied for plants and animals, but the ranks themselves were similar, with species as the basal rank followed by genus, family, order, class, phylum, and kingdom. With scant knowledge of the properties of viruses, it is no surprise that such a system was not immediately applied to these agents. Nevertheless, that human compulsion was too strong and, however flimsy the knowledge, there were initial attempts to classify the virus world. 

The first decades of virology, from the discovery of TMV until its purification by Wendell Stanley [7], are often referred to as the “biological” era (Figure 1), as the experiments that could be performed at that time provided information on the biological properties of a given virus. These properties included the hosts that a virus could infect and the symptoms it induced in each of these hosts (what is nowadays referred to as host range), as well as the way(s) in which it was naturally transmitted, (such as insect vectors or by seed, for plant viruses). This may seem hopelessly rudimentary by the standards of the 21st century, but the ingenuity of early 20th century virologists is not to be underestimated. Indeed, a simple assay developed for TMV allowed plant virologists to determine, among other things, that some plant viruses were comprised of more than one infectious unit (two, or even three). In modern parlance, they determined that the genomes of some plant viruses were divided into two or three components. The simple assay that allowed this discovery is known as the local lesion assay. Developed by Francis Holmes [8], the local lesion assay is based on the fact that TMV is not able to establish a systemic infection in the wild tobacco species *Nicotiana glutinosa*. The action of what is now known to be a single resistance gene [9] maintains the virus restricted to the initially infected cells, which undergo programmed cell death (also known as a hypersensitive response). Thus, a necrotic local lesion forms in the inoculated leaf. Holmes showed that the number of lesions was directly proportional to the inoculum concentration. The local lesion assay was therefore the first assay capable of quantifying the amount of virus in the host (albeit indirectly), which by itself would be a remarkable achievement (this was several years before the similar plaque assay was developed for bacteriophages). Other researchers, however, applying the same principle to other virus–host combinations, noticed that the relationship between inoculum dilution and the number of local lesions varied among different viruses. For TMV, it was a “one-hit curve”, while for cowpea mosaic virus (CPMV) it was a “two-hit” curve and for cucumber mosaic virus (CMV) it was a “three-hit” curve. In other words, the number of local lesions decreased more sharply for CMV compared to CPMV, and for CPMV compared to TMV, as the inoculum was diluted. Since Holmes had postulated that a local lesion was formed from the infection of a single cell, the conclusion followed that CPMV was comprised of two “infectious units” and CMV of three, while TMV had a single infectious unit—the sharper decrease being a consequence of the lower probability of two or three infectious units entering the same cell. Thus, from a very simple yet very elegant assay, a way of determining the number of genomic components of a plant virus was developed, which eventually became a taxonomic criterion (the history of the local lesion assay and its impact in plant virology is the topic of an excellent review by Scholthof [10]). 

It was perhaps inevitable that humans would attempt to classify viruses based on these biological properties. One of the earliest attempts was made in 1935, when Johnson and Hoggan proposed a set of descriptive keys based on five criteria: host, symptoms, mode of transmission, longevity in vitro, and thermal inactivation point [11]. The keys classified a set of 50 plant viruses into “groups”. Nevertheless, the continuous nature of biological properties makes them a poor set of criteria for taxonomy: the boundaries between categories are often blurred. For example, viruses with both DNA and RNA genomes may be transmitted by the same vector (begomoviruses and ipomoviruses, for example, are transmitted by the whitefly *Bemisia tabaci*). Moreover, they do not always reflect true evolutionary relationships, even in the case of the number of genomic components (many families have species with a different number of genomic components). As a result, these early taxonomic schemes were not adopted by the nascent virology community. The system proposed by Francis Holmes [12] to classify 89 plant viruses based on symptoms suffered the same fate. However, it is worth mentioning here due to its proposal for a binomial, Latinized nomenclature identical to the one used for other organisms. Thus, the genus *Marmor* would include all viruses which induced mosaic, and TMV would be named *Marmor tabaci* (although this nomenclature implies that TMV would be considered a species, the species concept was not used for viruses at the time). The fact that 53 of the 89 listed viruses belonged to the genus *Marmor* indicates the inadequacy of symptoms as a taxonomic criterion. As indicated above, the classification system was not adopted by the community. The binomial nomenclature system became collateral damage, and it would take almost 100 years for it to be implemented.

## 3. The Biochemical Era and the Creation of the International Committee on Taxonomy of Viruses

With the chemical purification of TMV in 1935 [7], the composition of viral particles could finally be determined. Although Stanley [7] proposed that TMV was composed solely of protein, Bawden et al. [13] performed a more detailed analysis of their purified preparations and detected the presence of a carbohydrate (ribose) and phosphorus, thus concluding that TMV was composed of protein and ribonucleic acid (RNA). At about the same time, the development of the transmission electron microscope finally allowed the visual observation of virus particles, and TMV was one of the first viruses to be “photographed” [14] (the first published images of viruses are of the ectromelia virus and the vaccinia virus; Von Borries et al. [15]). Thus, the “biochemical era” of virology started (Figure 1). From the 1940s to the 1970s, rapid developments in electron microscopy led to increasingly detailed observations of virus particle structure. X-ray diffraction and high-resolution/cryo-electron microscopy added a new dimension to our understanding of the viral particle architecture. The mid-1950s witnessed a number of major breakthroughs in the determination of viral particle structure and the function of its components. In 1955–1956, Rosalind Franklin and Donald Caspar resolved the structure of TMV [16,17,18], Fraenkel-Conrat and Williams showed that infectious TMV particles could be reassembled from purified RNA and protein [19], and Fraenkel-Conrat [20] and Gierer and Schramm [21] showed that RNA was the “infectious unit” of TMV (the role of viral nucleic acid as the infectious unit of viruses had been demonstrated by Hershey and Chase in 1952 for bacteriophage T2). Another major development of the biochemical era was represented by advances in serology, which led to better diagnostic methods. Subsequently, the amino acid sequence of the capsid protein of TMV was determined in 1960 [22,23], making it one of the first proteins to be sequenced (and the first viral protein). Thus, TMV continued to be at the forefront of virology. 

Taxonomy greatly benefited from the advances of the biochemical era. First and foremost, particle morphology and the nature of the viral nucleic acid (not only whether it was DNA or RNA, but also whether it was single- or double-stranded, and positive- or negative-sense in the case of RNA), as well as whether the genome was monopartite or segmented/multi-partite, quickly emerged as major taxonomic criteria (both used to this day). Many taxonomic systems were proposed during this time, but none were widely adopted. The ones that will be specifically mentioned here were proposed by Brandes and Wetter in 1959 and by Lwoff, Horne and Tournier in 1962. The system proposed by Brandes and Wetter [24] was based primarily on the particle morphology of elongated plant viruses, correlated with mode of transmission, thermal stability and serological relationships. The plant virus groups they proposed are still valid, and became genera such as *Tobamo*-, *Potex*-, *Carla*-, and *Potyvirus*. It was the first attempt to classify viruses for what they are, and not for the diseases they caused—a precursor of the presently accepted classification system. Like the one proposed by Holmes, it is singled out also because of its proposed nomenclature system—in English. In this system, TMV would be called tobacco mosaic virus (still not a species; it was referred to simply as a “virus”). This, of course, would be a strong departure from the Latinized binomial system adopted by other taxonomies. The system proposed by Lwoff et al. [25] was the most general ever devised up to that time, grouping plant, animal, and bacterial viruses in a Linnaean hierarchical scheme including the ranks of phylum, class, order, family, genus, and species. It was probably ahead of its time, and attracted criticism regarding the lack of information to justify the adoption of upper ranks and on the proposal of a binomial nomenclature. 

By the 1960s, the body of knowledge on viruses was significant, but a community-adopted taxonomy remained elusive. The need for a taxonomy, however, was acknowledged by most virologists. So it was that in 1966, during the 9th Congress of the International Association of Microbiology Societies held in Moscow, a group of virologists composed of the national representatives of several societies agreed on the creation of an International Committee on Nomenclature of Viruses (ICNV). The name was changed to the International Committee on Taxonomy of Viruses (ICTV) in 1975 to reflect the mandate of the committee (classification and nomenclature) more faithfully. A detailed history of the ICTV was published on the occasion of its 50th anniversary [26], and will not be repeated here. It should be mentioned that, at its very foundation, a number of principles were established that differentiated virus taxonomy from all other biological taxonomies: the committee would be in charge of both the creation and naming of taxa (in other taxonomies, only the names are determined by the corresponding committees); it was also determined that attempts would be made towards a Latinized nomenclature, although this would take quite a long time to finally be implemented. 

The establishment of the ICTV was instrumental in advancing virus taxonomy, even if community acceptance of early ICTV decisions was often contentious. Nevertheless, the most crucial aspect of virus taxonomy continued to be the definition of a set of criteria that accurately reflected the evolutionary relationships among viruses. The first report of the ICNV [27] established the first genera and families of viruses, and created several “groups” of apparently related viruses for which there was not enough evidence to allow the creation of taxa. It is of note that no genera or families were created for plant viruses, which were all listed as members of groups such as the “tobamovirus” group, which included TMV (the name being a combination of the first syllables of tobacco and mosaic, plus the word virus).

## 4. The Molecular Era and the Concept of Virus Species

During the late 1970s, molecular biology techniques became commonplace, and were quickly adopted by the virology community. With their small genomes, viruses were easily manipulated, and it did not take long for viral genomes to be completely sequenced [28]. More impressively, infectious cDNA clones of RNA viruses were developed, the first one for poliovirus [29]. In one of the rare instances in which TMV was not the object of a pioneer work in plant virology, the first infectious cDNA clone of a plant virus was constructed for the brome mosaic virus in 1984 [30]. Infectious clones of TMV were obtained shortly thereafter [31,32]. These advances brought about the “molecular virology” era (Figure 1). 

The availability of infectious clones allowed the application of reverse genetics to the study of viral gene function. Viral proteins could also be expressed in heterologous systems so that vast amounts of protein could be obtained, facilitating the analysis of their biochemical properties and facilitating immune assays. It was quickly demonstrated that all RNA viruses (regardless of host) encoded an RNA-dependent RNA polymerase, and that all plant viruses encoded a cell-to-cell movement protein. Although membranes from membrane-bound viruses (rhabdo-, bunya-, herpes-, poxviruses, etc.) are derived from host cells, proteins encoded by viruses are embedded in the membrane, and are important to determine host specificity, to initiate the infection process, and, in some cases, for vector transmission. Detailed studies of virus–host interactions were performed to elucidate attack and defense mechanisms, with the seminal discovery of post-transcriptional gene silencing (often referred to as RNA silencing) as the main defense mechanism against viruses in plants, fungi, and invertebrates [33]. TMV continued to be a protagonist in these studies, such as the pioneering studies of viral movement proteins using microinjection [34] and the production of the first transgenic plants expressing a viral protein (the TMV capsid protein) for pathogen-derived resistance [35].

If virus taxonomy benefited from the advances of the biochemical era, the discoveries that took place during the molecular era took it to a whole new level. Similarities in function among different viral proteins emerged as another useful taxonomic criterion [36,37,38]. However, it was the growing availability of genomic sequences that allowed virus taxonomy to take off, by facilitating the determination of phylogenetic relationships. As the output of Sanger sequencing machines increased (with the corresponding drop in cost), the availability of complete genome sequences became commonplace for most viruses. Sequence comparisons and phylogenetic relationships were quickly adopted as taxonomic criteria for many viruses [39,40,41,42]. These criteria were shown to be much more reliable than serological relationships, better reflecting the true evolutionary relationships among viruses. 

Simultaneous with, and partly as a consequence of, the great advances of the molecular era, the ICTV adopted the concept of virus species in its 7th Report [43]. From then on, species became the basal rank of virus taxonomy. The original definition of a viral species by van Regenmortel, adopted in the 7th Report, was changed in 2013. The current definition states that a species constitutes a monophyletic group of viruses whose properties can be distinguised from those of other species by multiple criteria. At present, most ICTV Study Groups require the availability of complete, or at least coding-complete, genomic sequences to create new species.

## 5. The Metagenomics Era—Unveiling the True Extent of the Virosphere

The technical limits of Sanger sequencing were reached in the late 20th century, allowing the routine sequencing of viral genomes and the widespread application of sequence-based analysis in virus taxonomy. For some viruses, such as HIV, thousands of isolates were sequenced, allowing in-depth evolutionary studies to be carried out. However, for most viruses, only one or a few isolates were sequenced. Some technical aspects of Sanger sequencing, such as the requirement of an oligonucleotide primer, limited its use as a virus discovery tool. In the early 21st century, new sequence technologies were developed that allowed the sequencing of billions (eventually, trillions) of nucleotides in a matter of days [44]. These high-throughput sequencing (HTS) technologies reduced the cost of sequencing eukaryotic genomes by several orders of magnitude, unleashing the genomics era. However, their greatest application in virology, the consequences of which are still reverberating, was in metagenomics. 

Metagenomics, defined as the “sequence analysis of environmental samples that contain an unknown mixture of life forms, often many that cannot be grown in culture” [45], was a sea change for virology and virus taxonomy. The sequence-unbiased nature of HTS technologies (which do not require molecular cloning or the use of an oligonucleotide primer) made them ideal tools for virus discovery [46]. Initial metagenomic studies immediately hinted at this potential, with a large percentage of sequences having no similarities to sequences in public databases. This so-called “viral dark matter” often constituted >50% of the sequences obtained. As more studies are performed (and therefore more sequences are deposited in the databases) and better bioinformatics tools are developed, the viral dark matter is slowly being illuminated. The massive number of new viruses identified in metagenomic studies has allowed major gaps in viral phylogenies to be filled, vastly advancing our understanding of the evolutionary relationships among viruses.

The impact of metagenomics in virus taxonomy cannot be overstated. The initial question was whether viruses identified solely in metagenomics studies should be classified at all. Of course, whole-genome sequence analysis has been instrumental for virus taxonomy since the molecular era, but a physical isolate was often required for a new species to be created. However, a physical isolate is often unavailable for viruses described in metagenomics studies. 

By the mid-2010s, the accumulation of metagenomic data posed a challenge to the ICTV. On one hand, incorporating such data would obviously improve the ability to infer evolutionary relationships, especially at the higher ranks. On the other, the lack of a physical isolate meant that in many cases basic biological information (such as host, transmission and pathogenicity, i.e., the fulfillment of Koch’s postulates) was absent. In 2016, the ICTV organized a workshop “to discuss frameworks for the advancement of virus taxonomy in the age of metagenomics” [47]. It was concluded that metagenomic sequences could be used in virus taxonomy as long as they satisfied a number of quality-control criteria. This was endorsed by the ICTV Executive Committee in its 2018 annual meeting [47].

Following this initial major step, several additional developments quickly followed. In 2019, the scope of virus taxonomy was expanded, with the creation of a 15-rank structure [48]. The previous structure encompassed only five ranks, with species as the basal rank and order at the top. The new structure added the primary ranks of class, phylum, kingdom and realm, as well as the corresponding secondary ranks (subclass, subphylum, etc.). The establishment of a 15-rank structure only became feasible with the incorporation of metagenomic data into the taxonomy scheme. Almost simultaneously, the first realm, *Riboviria*, was created to group all viruses with RNA genomes and encoding either an RdRp or a reverse transcriptase [49]. One year later, four new realms (as well as a large number of kingdoms, phyla, and classes) were created, effectively establishing a virus “megataxonomy” [50,51]. The latest release of the ICTV taxonomy, from March 2022, includes 6 realms, 10 kingdoms, 17 phyla, 39 classes, 65 orders, 233 families, 2606 genera and 10,434 species. As large as these numbers may seem, they most likely represent only a small fraction of the real diversity of the virosphere. The ICTV continues to face challenges regarding the way in which metagenomic sequences can be reliably classified. Nevertheless, our understanding of the high-level evolutionary relationships among viruses has dramatically increased thanks to the incorporation of metagenomics data [52,53], and will certainly continue to increase as new sequence analysis and bioinformatics tools are developed.

## 6. Tobacco Mosaic Virus: From *Contagium vivum fluidum* to *Riboviria*

Most of the new viruses that have been discovered by metagenomics do not seem to be pathogenic to their hosts [54], and the vast majority infect hosts other than plants [55,56,57,58]. Thus, even though TMV may have lost its protagonism in virus taxonomy during this latest era, the deeper understanding of virus evolutionary relationships brought about by metagenomics now allows for TMV to be classified up to the level of realm, as follows: species, *Tobacco mosaic virus*; genus, *Tobamovirus*; family, *Virgaviridae*; order, *Martellivirales*; class, *Alsuviricetes*; phylum, *Kitrinoviricota*; kingdom, *Orthornavirae*; realm, *Riboviria*. From *contagium vivum fluidum* to *Riboviria*, virus taxonomy owes many of its breakthroughs to TMV.

## Figures and Tables

**Figure 1 biomolecules-12-01363-f001:**
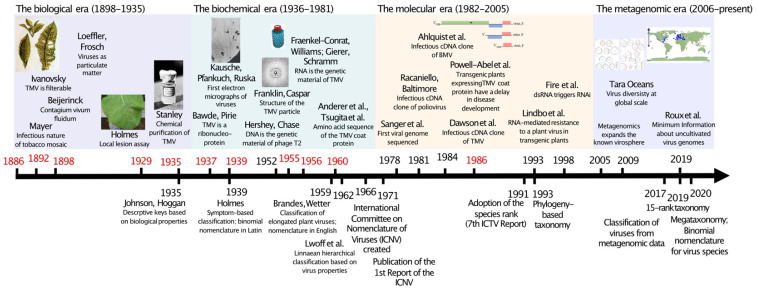
A timeline of virology (above the line) and virus taxonomy (below the line) milestones, from the discovery of TMV until the modern metagenomics era. Milestones involving TMV are indicated in red.

## Data Availability

Not applicable.

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
