# Peer review of "From Contagium vivum fluidum to Riboviria: A Tobacco Mosaic Virus-Centric History of Virus Taxonomy"

_biomolecules, 2022, doi:10.3390/biom12101363_

Round 1
Reviewer 1 Report
1. This review of virus taxonomy focuses on tobacco mosaic virus (TMV), the causative agent of the disease studied by Dmitry Ivanovsky, the first virus to be recognized, and often a major advance in virology throughout history in the 20th century, which is worthing learn from.
2. The significance of this review remains to be further elucidated.
Author Response
The reviewer made no specific comments, but states that the significance of the review remains to be determined. On this regard, we note that the manuscript was prepared following an invitation to contribute to the special issue commemorating Ivanovski's work on the discovery of tobacco mosaic virus. We had the idea to write a TMV-centric history of virus taxonomy and the idea was welcomed by the SI editor. It is our understanding that the final decision on the manuscript will be made by the editor based on a number of factors including its perceived significance, and we are ready to accept the editor's decision, whatever it may be.
Reviewer 2 Report
This paper gives a well written but brief overview of the history of virus taxonomy which, in the words of the authors centers around tobacco mosaic virus. Given it is relatively short it obviously cannot be exhaustive. It does give a nice short overview but especially the last part of the paper only mentions TMV very sporadically and centers much more around other viruses. This makes the title of the paper not factually correct.
More seriously however, unfortunately the paper does contain some serious factual mistakes which are indicated below. These, as well as a few minor points need to be addressed in a major revision before the paper is acceptable for publication.
Ln 42: Beyerinck initially performed these experiments in Wageningen, The Netherlands and definitely not in Belgium. Later he moved to Delft where he continued his work on what is now know to be TMV. In 1898 this
resulted in his publication describing the contagium vivum fluidum.
Ln 35: Yes, Adolf Mayer was German as his name suggests but he definitely performed these experiments in Wageningen, The Netherlands. And not in Germany. When investigating this disease of tobacco, (from 1876 onwards) which was first noticed and described in a few towns near Wageningen, he was the director of the Agricultural College in Wageningen which later became the Landbouwhogeschool and is now known as Wageningen University.
Ln 43: Ivanovsky never concluded that his ‘organism’ was not a bacterium and in fact concluded he was dealing with a microbe. Beyerinck however did conclude it could not be a bacterium and was the first to coin the term virus, simply indicating that whatever ‘unknown’ was causing the disease, was a ‘poison’.
Ln 55: the authors previously briefly explained the concept of the Chamberlain filter. They should do the same for the Kitasato filtratior. Especially since the paper they refer to will not be accessible for most readers.
Ln53: semantically the authors may seem correct but they seem to forget that the substance that was coined the contagium vivum fluidum was actually the plant sap containing the TMV. Beyerinck concluded that it had to contain something else than a fungus or bacterium. It was not until the 1930’s that by EM work the particle structure was established. Also most to all the plant virological work before that EM work was based on the infectious nature of plant sap.
Ln 107: could be or was? It is indeed a nice demonstration of clever science but the authors fail to show that at the time of discovery it was linked to taxonomy. In hindsight a lot of other clever experiments can be linked with taxonomy.
Some minor points of attention
Ln 100: the ‘official’ acronym for cowpea mosaic virus is CPMV
Ln 147: replace e with and?
Ln 161: the correct term is multi-partite instead of divided.
Ln 235: the authors could also mention that in certain systems these membrane-embedded viral proteins are even important in vector transmission
Ln 323: change ‘my’ to ‘by’
Author Response
L35, L42: Our sincere apologies for these geographical mistakes. Of course, both Mayer and Beijerinck worked in the Netherlands, not Germany/Belgium. Both mistakes have been corrected in the manuscript. On a side note, to the best of our knowledge, although Beijerinck started working on tobacco mosaic while in Wageningen, he performed the filtration work after moving to Delft. For example: Bos (Phil. Trans. R. Soc. Lond. B 354, 675-685, 1999) states (p. 677) that "In 1895, at the age of 45, the more academic phase of Beijerinck's career began when he was appointed Professor of Bacteriology at the Polytechnical School (now Technical University) at Delft. When, two years later (in 1897), a new bacteriology laboratory and greenhouse were built, he immediately commenced the series of decisive experiments that would lead to the classic but preliminary paper read before the Royal Academy on the 26th of November 1898 (Beijerinck 1898a)". But this is a moot point, as both Wageningen and Delft are in the Netherlands...
L43: The reviewer is absolutely correct that Ivanovski never concluded that the causal agent of tobacco mosaic was not a bacterium. However, we never made this claim in the manuscript ! Our text in L43 states that "Beijerinck concluded that the causal agent of tobacco mosaic was not a fungus or a bacterium".
L53: This is a very interesting comment, and the reviewer's point is well taken. But we're going to cite Bos again, who states on p. 678 of the same paper mentioned above that "Beijerinck's conclusion, therefore, was that infection is not due to a microbe (a contagium fixum; Beijerinck 1898a), but to a non-corpuscular (that is, non-cellular) entity which he named contagium vivum fluidum. In the three versions of his paper the terms 'liquid state' and 'dissolved state' are used interchangeably (Beijerinck 1898a,b, 1900a)". The title of the 1900 publication is "De l'existence d'un principe contagieux vivant fluide, agent de la nielle des feuilles de tabac". We certainly agree with the reviewer that most, if not all, of the plant virology work performed until the development of the TEM was based on the infectious nature of plant sap, but it is clear that Beijerinck originally used "fluidum" to refer to the agent itself. Additional accounts by Murphy (Adv. Virus Res. 95:197-220, 2016) and Mettenleiter (Adv. Virus Res. 99:1-16, 2917) agree with this conclusion. Thus, we would like to leave the text unchanged.
L55: Good suggestion - done.
L107: We thank the reviewer for this comment. It did become a taxonomic criterion, although much later. The text was changed to better express this.
Ln 100: Actually, there's no such thing as an official acronym, as acronyms have no taxonomic value and thus are not regulated by the ICTV. But we agree with the reviewer that CPMV is the most used form in the literature. Thus, corrected.
Ln 147: Sorry, EndNote configuration mistake. Corrected.
Ln 161: Corrected.
Ln 235: Good suggestion - Done.
Ln 323: Corrected.
Round 2
Reviewer 2 Report
The authors have dealt adequately with my comments